# Clinical Characteristics of Punctate Hyperfluorescence Spots in the Fellow Eye of Patients with Unilateral Macular Neovascularization with No Drusen

**DOI:** 10.3390/jcm13185394

**Published:** 2024-09-12

**Authors:** Hiroyuki Kamao, Katsutoshi Goto, Yuto Date, Ryutaro Hiraki, Kenichi Mizukawa, Atsushi Miki

**Affiliations:** 1Department of Ophthalmology, Kawasaki Medical School, 577 Matsushima, Kurashiki 701-0114, Okayama, Japan; k_goto@med.kawasaki-m.ac.jp (K.G.); theethee1103@gmail.com (Y.D.); hiraki@med.kawasaki-m.ac.jp (R.H.); amiki@med.kawasaki-m.ac.jp (A.M.); 2Shirai Eye Hospital, 1339 Takasecho Kamitakase, Mitoyo 767-0001, Kagawa, Japan; sorararaharururu@gmail.com

**Keywords:** pachychoroid neovasculopathy, punctate hyperfluorescence spot, no-drusen, anti-vascular endothelial growth factor therapy

## Abstract

**Objectives**: To assess the clinical characteristics of patients with macular neovascularization (MNV) with no drusen in the fellow eye, we investigated the incidence of MNV in fellow eyes and the outcomes of intravitreal aflibercept (IVA) monotherapy in MNV eyes of patients with unilateral MNV with a punctate hyperfluorescence spot (PHS) in the fellow eye. **Methods**: We retrospectively studied 58 treatment-naïve patients with unilateral MNV with no drusen in the fellow eye. Patients were classified into a PHS group (*n* = 29) or no-PHS group (*n* = 29) based on the presence of PHS. We evaluated the incidence of MNV in the fellow eye, and the retreatment rate after initiation of three monthly aflibercept injections over one year. **Results**: Fellow eyes in the PHS group had a thicker choroid (*p* < 0.05) and higher prevalence of pachychoroid pigment epitheliopathy (PPE) (*p* < 0.001). MNV eyes in the PHS group had a thicker choroid (*p* = 0.09). The PHS group had a lower retreatment rate (*p* < 0.05) and required fewer injections (*p* < 0.05) than the no-PHS group. MNV developed in one eye in both the PHS and no-PHS groups, and both cases occurred in areas of hypofluorescence on indocyanine green angiography within the PPE area before the onset of MNV. **Conclusions**: The PHS group frequently exhibited pachychoroid disease characteristics and responded better to IVA monotherapy than the no-PHS group. These groups may represent distinct populations of patients with unilateral MNV with no drusen in the fellow eye.

## 1. Introduction

Macular neovascularization (MNV), secondary to age-related macular degeneration (AMD), is a leading cause of blindness in elderly populations in developed countries. Although anti-vascular endothelial growth factor (VEGF) therapy has improved visual outcomes in patients with MNV, it requires high costs and regular visits to medical institutions; therefore, it is important to identify predictive factors for the efficacy of anti-VEGF therapy to reduce the treatment burden.

Freund et al. proposed a new disease concept called pachychoroid, and this concept has classified MNV secondary to AMD into pachychoroid-driven MNV and drusen-driven MNV [1,2]. Pachychoroid-driven MNV, named pachychoroid neovasculopathy (PNV), is characterized by MNV associated with a pathologically thick choroid (pachychoroid) [2]. Pachychoroid diseases include several clinical entities, such as PNV, central serous chorioretinopathy (CSC) [3], pachychoroid pigment epitheliopathy (PPE) [1], polypoidal choroidal vasculopathy (PCV) [4], peripapillary pachychoroid syndrome [5], and focal choroidal excavation [6]. In particular, CSC, PPE, PNV, and PCV have been speculated to represent a sequential disease process caused by a pathologically thick choroid [7]. Although the diagnostic criteria of PNV are not established, it is characterized by dilated choroidal vessels (pachyvessels) in Haller’s layer, attenuated choriocapillaris and choroidal vessels in Sattler’s layer [4], choroidal vascular hyperpermeability (CVH) [8], and absence of typical drusen, except for pachydrusen [9]. CVH, first described in CSC, is a common feature in pachychoroid diseases. Before the pachychoroid concept was established, features such as choroidal filling delay and venous dilation were frequently detected in eyes with CSC [10], making it a representative condition of pachychoroid diseases.

The clinical characteristics of eyes with PNV include the absence of drusen or pachydrusen. However, studies that classified the fellow eye of patients with unilateral MNV according to the type of drusen reported that the pachydrusen group required less intravitreal aflibercept (IVA) monotherapy than the no-drusen group [11,12]. Therefore, patients with unilateral MNV with pachydrusen in the fellow eye and those with no drusen in the fellow eye may comprise distinct populations. Pachydrusen is a relatively new subtype of drusen associated with the pachychoroid [9]. However, there have been no reports focusing on the clinical characteristics of patients with unilateral MNV with no drusen in the fellow eye.

A punctate hyperfluorescence spot (PHS) is a focal area of hyperfluorescence observed on mid- to late-phase indocyanine green angiography (ICGA) in most eyes with CSC and PCV [13,14]. Large-sized PHS is a clinical feature of pachydrusen and clearly distinguishes pachydrusen from soft drusen, which shows hypofluorescence on ICGA [11]. In this study, patients with unilateral MNV patients with no significant drusen (no drusen) in the fellow eye were classified into two groups based on the presence of PHS (PHS group) and the absence of PHS (no-PHS group) in the fellow eye. We investigated the clinical characteristics of the patients, focusing primarily on their response to IVA monotherapy.

## 2. Materials and Methods

### 2.1. Study Design

We retrospectively studied 119 consecutive treatment-naïve patients with unilateral MNV with no drusen in the fellow eye who were treated with a pro re nata (PRN) regimen at Kawasaki Medical School between January 2016 and December 2023. Data on hypertension, diabetes, and cigarette smoking were collected from hospital or patient records. Patients were classified into never-smokers and ever-smokers, as described in a previous report [15].

A previous study reported that one third of CSC patients with flat irregular PED had MNV [16]; therefore, patients with MNV having flat irregular PED in the fellow eye were classified as patients with bilateral MNV and were excluded from the present study. We excluded patients who had received anti-VEGF agents other than aflibercept (bevacizumab, pegaptanib, ranibizumab, brolucizumab, and faricimab) or had undergone verteporfin photodynamic therapy, laser photocoagulation, or vitrectomy. We also excluded patients with MNV because of high myopia (>−6 diopters), uveitis, or angioid streaks, and patients with eye diseases that could potentially influence the treatment outcome of the studied eye, such as branch retinal vein occlusion, diabetic retinopathy, or glaucoma.

No-drusen cases were determined using fundus color photography, near-infrared en face, spectral domain optical coherence tomography (OCT) and swept-source OCT according to the criteria presented in previous studies [12,17]. Briefly, pachydrusen was defined as yellow-white deposits on color fundus photographs corresponding to sub-RPE accumulation on OCT images. An OCT scan of the pachydrusen showed drusenoid deposits above the pachyvessels. Pachydrusen showed a hyperfluorescent area during the mid- to late-phase ICGA. Among drusen showing hyperfluorescence on the ICGA, those smaller than 125 µm in size were classified as no-drusen.

Among the 119 eyes of the 119 consecutive patients with unilateral MNV with no drusen in the fellow eye, we excluded 61 eyes of 61 patients: 31 lacked ICGA data, 18 were treated with different methods, and 12 were followed up for less than one year. We classified 58 patients with MNV into two groups based on the presence (Figure 1) and absence (Figure 2) of PHS in the fellow eyes. PHS was investigated in all fellow eyes using mid- to late-phase ICGA based on previous studies [13,14]. The distribution of PHS was typically scattered as single or clustered hyperfluorescent spots.

PPE was diagnosed when all conditions were present out of the following: (1) abnormalities of the retinal pigment epithelium on fundus autofluorescence (FAF) or fluorescence angiography images without past or current subretinal fluid, such as the presence of a descending tract; and (2) abnormalities of the retinal pigment epithelium accompanied by pachyvessels confirmed by OCT and ICGA images.

We classified the subretinal hyper-reflective material (SHRM), hyper-reflective signal above the retinal pigment epithelium on OCT images, into six types using fundus color photography, FAF, swept-source OCT, OCT angiography, FA, and ICGA according to the following criteria presented in previous studies [18]: exudation, hemorrhage, neovascular tissue, vitelliform, fibrosis, and no SHRM (Figure 3). The exudate type was defined as yellow-white aggregates on color fundus photographs, corresponding to hypofluorescence on early-phase FA and ICGA images. The hemorrhage type was defined as red aggregates on color fundus photographs, corresponding to hypofluorescence on FA and ICGA images. The neovascular tissue type was defined as yellow-white aggregates on color fundus photographs, corresponding to hyperfluorescence on early-phase FA and ICGA images and hyperfluorescence (leakage) on late-phase FA images. The vitelliform type was defined as yellow aggregates on color fundus photographs, corresponding to hyperfluorescence on FAF images. The fibrosis type was defined as yellow aggregates on color fundus photographs, corresponding to hyperfluorescence (staining) on late-phase FA images. None of eyes in this study had the vitelliform or fibrosis type.

### 2.2. Treatment Method

All enrolled patients received three monthly injections (loading dose regimen) of aflibercept and were followed up monthly as the PRN regimen. If retinal fluid (SRF, subretinal fluid and/or IRF, intraretinal fluid) was detected on OCT images after the loading dose regimen, all eyes with retinal fluid were treated using the treat-and-extend protocol of IVA therapy. The treatment interval, which was adjusted for 2 to 4 weeks, was reduced when the retinal exudate recurred or persisted and extended when no retinal exudate occurred for six months.

### 2.3. Research and Analysis

All enrolled participants underwent a complete ophthalmologic examination, including measurements of best-corrected visual acuity (BCVA), indirect ophthalmoscopy, slit-lamp biomicroscopy with a noncontact lens, color fundus photography and FAF (Canon CX-1; Canon, Tokyo, Japan), spectral domain OCT (RS-3000 Advance OCT; Nidek Corporation, Gamagori, Japan), swept-source OCT (DRI OCT-1 Atlantis; Topcon Corporation, Tokyo, Japan), and fluorescein and ICGA (HRA-2; Heidelberg Engineering GmbH, Dossenheim, Germany). Visual acuity data were obtained as decimal visual acuity (BCVA) values and converted to the logarithm of the minimum angle of resolution (logMAR) units for analysis. Central retinal thickness (CRT) and subfoveal choroidal thickness (SFCT) were measured using swept-source OCT, as previously described [15]. We measured BCVA, CRT, and SFCT in eyes with MNV at baseline and 12 months after the initial treatment to determine the one-year outcome of aflibercept monotherapy. We assessed the incidence of retinal exudate after the loading dose regimen in eyes with MNV and the progression to MNV in the fellow eye.

### 2.4. Statistical Analysis

Statistical analyses were performed using the JMP Pro 17 software (SAS Institute, Cary, NC, USA). Age, BCVA, CRT, SFCT, and the number of aflibercept injections received were compared between the two groups using the Kruskal–Wallis test followed by the Steel–Dwass test. Pearson’s chi-squared test was used to compare the differences in the female-to-male ratio; incidence of hypertension, diabetes, and ever-smokers; presence of IRF, SRF, and polypoidal lesions in the MNV eye; SHRM subtype ratio in the MNV eye; proportion of dry macula after the loading dose regimen in the MNV eye; and presence of PPE in the fellow eye. The one-year incidences of retreatment in the MNV eye were constructed using the Kaplan–Meier estimator. A log-rank test was used to compare differences among the four groups. Statistical significance was set at *p* < 0.05.

## 3. Results

### 3.1. Patient Characteristics

A total of 58 eyes from 58 patients with treatment-naïve unilateral MNV were enrolled in our study. These patients were classified into two groups according to the presence or absence of PHS (Figure 1G) in the fellow eye, and their baseline characteristics are presented in Table 1. The analysis included 29 patients (9 women, 20 men; mean [±SD] age 69.8 ± 9.6 [range, 52–91] years) in the PHS group and 29 patients (8 women, 21 men; mean age 73.2 ± 8.2 [range, 52–87] years) in the no-PHS group. The mean SFCTs of the fellow eyes at the baseline were 249.9 ± 124.4 µm and 187.0 ± 84.8 µm in the PHS and no-PHS groups, respectively. The PHS group had a thicker SFCT than the no-PHS group (*p* = 0.048). The proportion of patients with PPE was 79.3% and 37.9% in the PHS and no-PHS groups, respectively, with a higher proportion of PPE in the PHS group (*p* < 0.001). The proportions of patients with exudation, hemorrhage, neovascular tissue, and no SHRM were 48.3%, 27.6%, 0.0%, and 24.1%, respectively, in the PHS group, and 20.7%, 24.1%, 6.9%, and 48.3%, respectively, in the no-PHS group. The PHS group had a higher proportion of exudation type of SHRM (*p* = 0.036). The two study groups were comparable in terms of age, female-to-male ratio, and proportion of patients with hypertension, diabetes, IRF, SRF, and polypoidal lesions in the MNV eyes.

### 3.2. One-Year Outcome of IVA Therapy

A total of 29 eyes in the PHS group and 29 eyes in the no-PHS group were followed up for at least one year after the loading dose regimen with aflibercept (Table 2). The mean BCVA, CRT, and SFCT of the eyes with MNV at the baseline were 0.35 ± 0.36, 317.8 ± 97.2 µm, and 267.2 ± 120.4 µm, respectively, in the PHS group, and 0.23 ± 0.20, 318.7 ± 60.6 µm, and 213.8 ± 94.0 µm, respectively, in the no-PHS group. There were no significant differences in BCVA or CRT at baseline (BCVA, *p* = 0.21; CRT, *p* = 0.41); however, the PHS group had a thicker choroid (SFCT, *p* = 0.09).

The mean BCVA, CRT, and SFCT of the eyes with MNV at 12 months after initial treatment were 0.13 ± 0.29, 212.6 ± 38.1 µm, and 231.6 ± 129.1 µm, respectively, in the PHS group, and 0.10 ± 0.18, 253.4 ± 69.2 µm, and 188.8 ± 94.6 µm, respectively, in the no-PHS group. There were no significant differences in BCVA or SFCT (BCVA, *p* = 0.65; SFCT, *p* = 0.36), and the no-PHS group had a thicker CRT than the PHS group (*p* < 0.004). The proportion of dry macula cases after the loading dose regimen was 79.3% and 62.1% in the PHS and no-PHS groups, respectively (*p* = 0.15). The mean number of aflibercept injections was 5.4 ± 2.2 and 7.0 ± 2.9 in the PHS and no-PHS groups, respectively (*p* = 0.039). The two study groups were comparable in terms of the proportion of dry macula cases after the loading dose regimen, and the no-PHS group required frequent aflibercept injections.

The retreatment rates 12 months after the loading dose regimen were 65.5% (19/29) and 86.2% (25/29) in the PHS and no-PHS groups, respectively. The Kaplan–Meier curve for the retreatment proportion differed significantly between the two groups (*p* = 0.042). The PHS group had a lower retreatment rate than that of the no-PHS group (Figure 4).

### 3.3. Incidence of MNV in the Fellow Eye

We evaluated 29 eyes each in the PHS group and 29 eyes in the no-PHS group which were followed up for over 12 months after the initial treatment for the incidence of MNV in the fellow eye. One eye in each group developed MNV in the fellow eye during the mean follow-up duration of 67.0 ± 53.3 months (14.3–170.6 months). One eye in the PHS group developed MNV as PNV in the fellow eye 25 months after the initial treatment, and one eye in the no-PHS group developed MNV as PCV in the fellow eye 69 months after the initial treatment (Figure 5 and Figure 6). Both MNVs developed in areas of hypofluorescence adjacent to choroidal vascular dilation on ICGA images within areas of PPE before the onset of MNV.

## 4. Discussion

Our study revealed that in patients with unilateral MNV with no drusen in the fellow eye, fellow eyes in the PHS group had a thicker choroid and a higher prevalence of PPE. MNV eyes in the PHS group had a thicker choroid and a higher prevalence of exudate-type of SHRM. Tsujikawa et al. reported that PHS detected by mid- to late-phase ICGA was observed in 93% (38/41 eyes) of eyes with CSC and 78% (29/37 eyes) of fellow eyes in patients with CSC [13]. Park et al. investigated the relationship between the incidence of PHS and diseases, including typical AMD, CSC, and PCV [14]. Eyes with pachychoroid diseases, including CSC and PNV, showed a thicker choroid and a higher prevalence of PHS than those with typical AMD. Although the study cohorts differed, our findings were consistent with those of previous studies.

However, the pathogenesis of PHS is not fully understood. Previous studies have suggested that CVH is expanded from the PHS localized in the inner choroid [13], and some PHSs show late staining of drusen-like subretinal pigment epithelium deposits associated with CVH [19]. A previous study by Kang et al. showed that pachydrusen is strongly associated with CVH and PHS in patients with PCV and PNV [20]. In this study, the higher prevalence of exudate-type SHRM in the PHS group might have resulted from CVH. Therefore, PHS could be an accumulation of exudate under the retinal pigment epithelium due to CVH, and a large PHS is diagnosed as pachydrusen. A previous study demonstrated that patients with PCV who received IVA therapy were classified into four groups based on the drusen type in the fellow eye: the pachydrusen group, no-drusen group, soft drusen group, and PCV/scarring group [11]. The retreatment-free period was longer in the pachydrusen group than in the other groups. We previously classified patients with MNV into four groups based on the type of drusen in the fellow eye: the pachydrusen, no-drusen, soft drusen, and SDD groups [12]. The pachydrusen group had a longer retreatment-free period than the other groups. Considering the PHS and no-PHS groups in this study as the pachydrusen and no-drusen groups, respectively, the response to IVA monotherapy was consistent with previous results. The effectiveness of anti-VEGF therapy varies depending on the pathogenesis and patient age. In the pathogenesis of MNV, anti-VEGF therapy is more effective in patients with PNV than in those with neovascular AMD (nAMD) [21,22]. In patients with MNV, older age was associated with recurrence after the loading dose regimen of anti-VEGF therapy [23]. In this study, there was no difference in age between the two groups; therefore, the lower retreatment proportion in the PHS group may have resulted from a difference in the mechanism underlying the development of MNV. Kuranami et al. reclassified patients with MNV previously categorized using conventional methods as typical AMD, PCV, retinal angiomatous proliferation (RAP), into PNV and non-PNV, using a new method [24]. They showed that 45% (17/38 eyes) of patients with PCV (MNV with polypoidal lesions) were reclassified into the PNV group, and 55% (21/38 eyes) of patients with PCV were reclassified into the non-PNV group, indicating that polypoidal lesions are not a hallmark of pachychoroid disease. Drusen is a hallmark of AMD, although the percentage of patients with drusen was 19.3% in the non-PNV group in their study. They suggested that there might be a mechanism for the development of MNV that is neither pachychoroid- nor drusen-driven. In this study, the no-PHS group had no drusen and lacked characteristic pachychoroid findings, such as PPE or a thicker choroid; however, the prevalence of polypoidal lesions was not significantly different between the PHS and no-PHS groups. Therefore, the no-PHS group may include many patients with MNV that is neither pachychoroid- nor drusen-derived.

In the pathogenesis of MNV, PNV is considered an ischemic disease resulting from dilated choroidal vessels or pachyvessels compressing the immediate overlying choriocapillaris, leading to RPE dysfunction and the development of MNV [25]. The present study showed that PCV developed in one patient in each of the PHS and no-PHS groups. In both cases, the MNV developed in the areas of choroidal filling defects within the PPE area before the onset of the MNV. In a previous study, reduced choriocapillary flow density and increased choroidal thickness were colocalized in the CVH area in eyes with PPE, suggesting that choroidal ischemia is associated with the pathogenesis of PPE [26]. It has been speculated that CSC, PPE, PNV, and PCV represent a continuum of disease processes caused by persistent choroidal dysfunction [27]. However, because PPE is asymptomatic, reports on the continuous progression of pachychoroid diseases, including the development of MNV from PPE, are limited. Yagi et al. reported that PNV developed in one of 148 eyes with PPE during a follow-up period of 46.4 months [28]. Tang et al. reported seven cases of PCV developed from PPE; however, most eyes with PPE had flat irregular PED at baseline [29]. A previous report demonstrated that flat irregular PED sometimes includes MNV, even in the absence of a retinal exudate [16]. In this study, unilateral MNV patients with irregular PED in the fellow eye were excluded, and neither eye in which MNV developed from PPE showed flat irregular PED before the onset of MNV. Our results have important clinical implications because PNV and PCV developed in areas of PPE, particularly choroidal ischemia, which showed hypofluorescence on ICGA images.

MNVs are the leading cause of blindness in the elderly populations of developed countries. Anti-VEGF therapy has improved visual outcomes in patients with MNV; however, expensive anti-VEGF agents place a heavy burden on these patients. Therefore, identifying predictive factors for the efficacy of anti-VEGF therapy in patients with MNV is crucial. Anti-VEGF therapy is more effective in patients with PNV than in those with nAMD [21,22]; therefore, accurate diagnosis of PNV and nAMD is important. The diagnostic criteria delineating between PNV and nAMD have not been established because choroidal thickness, which is correlated with patient age and axial length [30,31], has no definitive cutoff value. Our study revealed that the PHS group had higher choroidal thickness in the eyes with MNV as well as fellow eyes, a higher prevalence of PPE, and a lower proportion of patients retreated with IVA therapy. PHS is strongly associated with pachydrusen, and PPE is a pachychoroid disease. Therefore, the PHS group of patients with unilateral MNV with no drusen in the fellow eye could be diagnosed with PNV.

The limitations of this study include the relatively small sample size and its retrospective nature. Particularly in studies on the incidence of MNV in fellow eyes, the follow-up interval was not standardized; therefore, areas of MNV development were not checked regularly. Moreover, patients with flat irregular PED in the fellow eye were excluded; however, we did not evaluate all areas using OCT. Lee et al. reported that quiescent PNVs are asymptomatic for prolonged periods, without retinal exudate [32]. Therefore, we cannot eliminate the possibility that the fellow eyes had flat irregular PED, including MNV. The no-PHS group in this study had no drusen and lacked pachychoroid disease characteristics (thinner choroid, lower prevalence of PPE, and frequent anti-VEGF therapy). There may be a mechanism for the development of MNV that is not caused by pachychoroid or drusen; however, this is currently unknown, and further research on the mechanism of MNV development is needed.

## 5. Conclusions

Among patients with unilateral MNV with no drusen in the fellow eye, those with PHS in the fellow eye had a thicker choroid, higher prevalence of PPE, decreased retinal thickness, and a lower percentage of patients retreated with IVA therapy than those with no-PHS in the fellow eye. This study showed that the PHS group had many clinical features of pachychoroid disease. Hence, patients with unilateral MNV, with and without PHS in the fellow eye, may represent distinct patient populations.

## Figures and Tables

**Figure 1 jcm-13-05394-f001:**
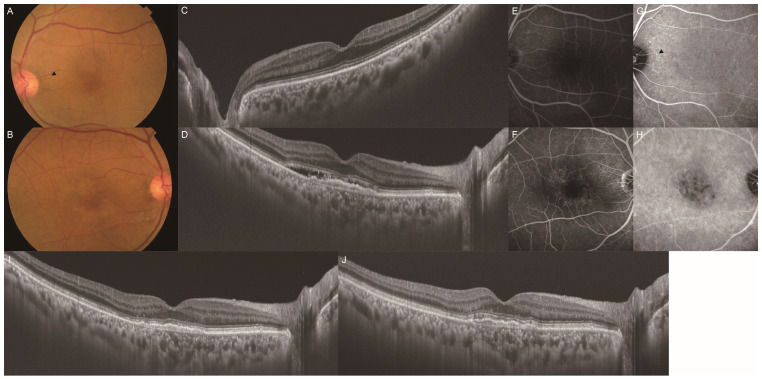
A representative case of unilateral MNV with PHS in the fellow eye. Images (**A**,**C**,**E**,**G**) depict the fellow eye, and images (**B**,**D**,**F**,**H**) depict the MNV eye. (**A**) Color fundus photography revealed no drusen greater than 125 µm in size. Small yellowish drusenoid deposits (arrowhead) were observed around the temporal optic disc. (**B**) Color fundus photography revealed yellowish lesions in the macula. (**C**) B-scan with a swept-source OCT image of the macula showed no retinal exudate. (**D**) B-scan with a swept-source OCT image of the macula showed fibrovascular PED with subretinal fluid. (**E**) Late-phase fluorescein angiography (FA) imaging showed no obvious abnormality. (**F**) Late-phase FA imaging showed an area of hyperfluorescence corresponding to yellowish lesions in the macula. (**G**) Late-phase ICGA imaging showed PHS (arrowhead) corresponding to yellowish deposits around the temporal optic disc. (**H**) Late-phase ICGA imaging showed multiple areas of hyperfluorescence and hypofluorescence corresponding to hyperfluorescence on FA image. (**I**) B-scan with a swept-source OCT image of the macula after the loading dose regimen showed no retinal exudate. (**J**) B-scan with a swept-source OCT image of the macula 1 year after the initial administration of aflibercept showed no retinal exudate. This patient received 3 injections for 1 year.

**Figure 2 jcm-13-05394-f002:**
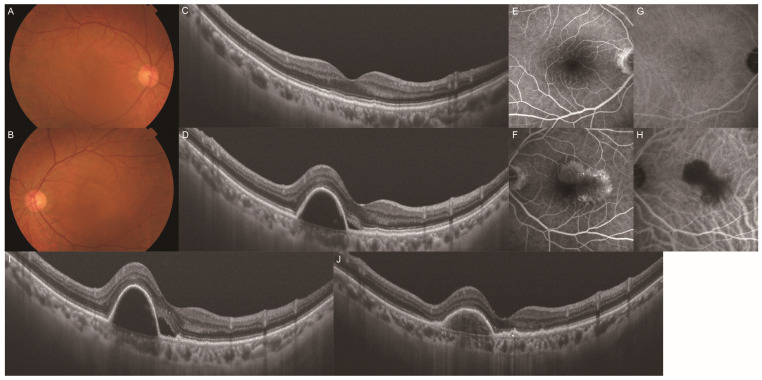
A representative case of unilateral MNV without PHS (no-PHS) in the fellow eye. Images (**A**,**C**,**E**,**G**) depict the fellow eye, and images (**B**,**D**,**F**,**H**) depict the MNV eye. (**A**) Color fundus photography showed no obvious abnormality. (**B**) Color fundus photography revealed color changes in the macula. (**C**) B-scan with a swept-source OCT image of the macula showed no retinal exudate. (**D**) B-scan with a swept-source OCT image of the macula showed serous PED with subretinal fluid. (**E**) Late-phase fluorescein angiography (FA) imaging showed no obvious abnormality. (**F**) Late-phase FA imaging showed areas of hyperfluorescence in the macula. (**G**) Late-phase ICGA imaging showed no obvious abnormality. (**H**) Late-phase ICGA imaging showed areas of hypofluorescence corresponding to serous PED. (**I**) B-scan with a swept-source OCT image of the macula after the loading dose regimen showed subretinal fluid. The treatment method for this patient was changed from the PRN protocol to the treat-and-extend protocol. (**J**) B-scan with a swept-source OCT image of the macula 1 year after the initial administration of aflibercept showed no retinal exudate. This patient received monthly injections after the loading dose regimen.

**Figure 3 jcm-13-05394-f003:**
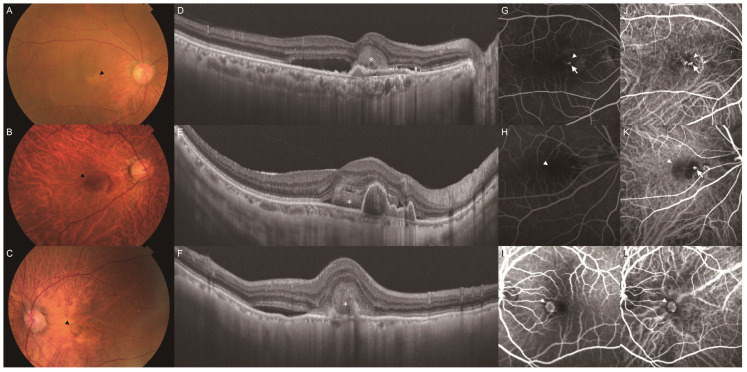
Representative cases of SHRM in the MNV eye. Images (**A**,**D**,**G**,**J**) represent the exudation type, images (**B**,**E**,**H**,**K**) represent the hemorrhage type, and images (**C**,**F**,**I**,**L**) represent the neovascular tissue type. (**A**) Color fundus photography revealed yellow-white aggregates (arrowhead) in the macula. (**B**) Color fundus photography revealed red aggregates (arrowhead) in the macula. (**C**) Color fundus photography revealed yellow aggregate (arrowhead) in the macula. (**D**) B-scan with a swept-source OCT image of the macula showed an SHRM (asterisk) above the fibrovascular PED. (**E**) B-scan with swept-source OCT image of the macula showed an SHRM (asterisk) next to the subretinal pigment epithelial hemorrhage. (**F**) B-scan with a swept-source OCT image of the macula showed an SHRM (asterisk) above the retinal pigment epithelium. The hyper-reflective line of retinal pigment epithelium was attenuated. (**G**) Early-phase FA imaging showed areas of hypofluorescence (arrowhead) corresponding to yellow-white aggregates and areas of hyperfluorescence (arrow) corresponding to MNV on ICGA images. (**H**) Early-phase FA imaging showed an area of hypofluorescence (arrowhead) corresponding to red aggregates. (**I**) Early-phase FA imaging showed an area of hyperfluorescence (arrowhead) corresponding to yellow aggregates. (**J**) Early-phase ICGA imaging showed an area of hypofluorescence (arrowhead) corresponding to yellow-white aggregates and an area of hyperfluorescence (arrow) as MNV. (**K**) Early-phase ICGA imaging showed an area of hypofluorescence (arrowhead) corresponding to red aggregates and an area of hyperfluorescence (arrow) as a polypoidal lesion. (**L**) Early-phase ICGA imaging showed an area of hyperfluorescence (arrowhead) as MNV.

**Figure 4 jcm-13-05394-f004:**
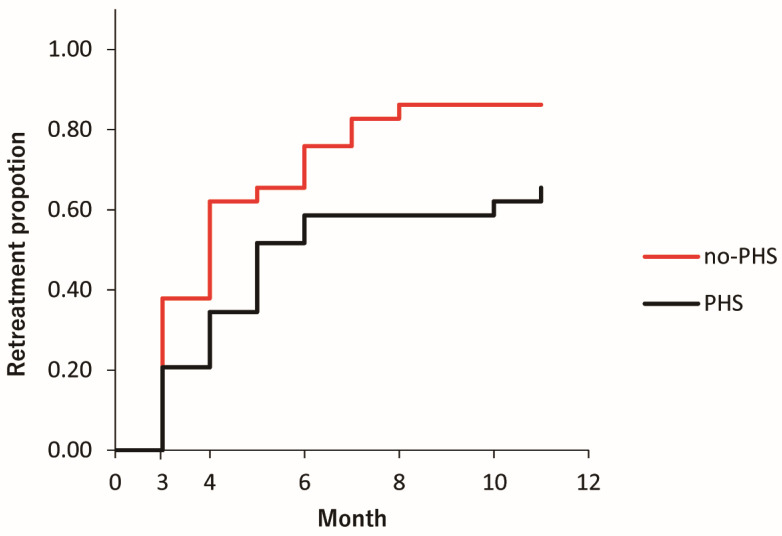
Kaplan–Meier curves for the retreatment proportions after the loading dose regimens. The black line represents the PHS group, and the red line represents the no-PHS group. Month 3 represents one month after the loading dose regimen.

**Figure 5 jcm-13-05394-f005:**
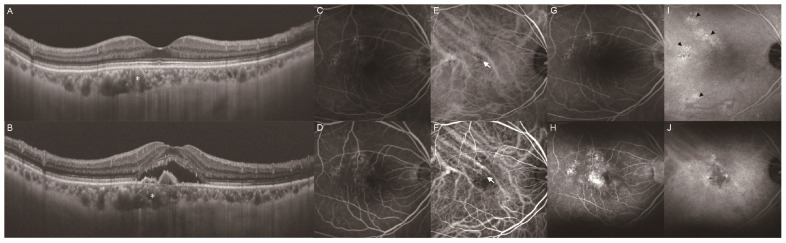
Development of MNV in the fellow eye of a patient with PHS in the fellow eye. Images (**A**,**C**,**E**,**G**,**I**) represent 25 months before the development of MNV in the fellow eye. Images (**B**,**D**,**F**,**H**,**J**) represent the onset of MNV in the fellow eye. (**A**) B-scan with a swept-source OCT image of the macula showed pachyvessels (asterisk) without PED in the superior fovea. (**B**) B-scan with a swept-source OCT image of the macula showed PED with SRF above the pachyvessels (asterisk). (**C**,**D**) Early-phase FA imaging showed the area of hyperfluorescence in the temporal superior macula. (**E**) Early-phase ICGA imaging showed pachyvessels corresponding to the area of hyperfluorescence on FA image. An area of hypofluorescence (arrow) was observed adjacent to choroidal vascular dilation. (**F**) Early-phase ICGA imaging showed abnormal vessels (arrow) in the area of hypofluorescence on ICGA prior to the onset of MNV. (**G**) Late-phase FA imaging showed an area of hyperfluorescence in the temporal superior macula. (**H**) Late-phase FA imaging showed areas of hyperfluorescence in the temporal and temporal superior macula. (**I**) Late-phase ICGA imaging showed PHS with CVH (arrowhead) in the temporal superior macula corresponding to the area of hyperfluorescence on the FA image. (**J**) Late-phase ICGA imaging showed areas of hyperfluorescence and hypofluorescence in the macula.

**Figure 6 jcm-13-05394-f006:**
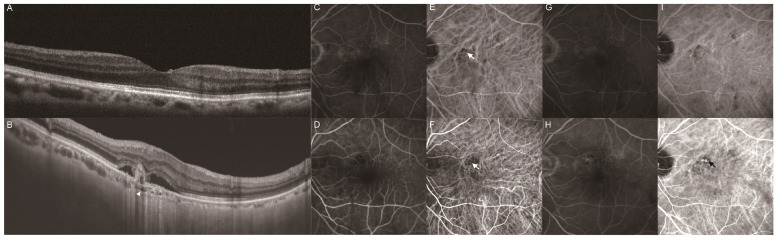
Development of MNV in the fellow eye of a patient without PHS in the fellow eye. Images (**A**,**C**,**E**,**G**,**I**) represent 69 months before the development of MNV in the fellow eye. Images (**B**,**D**,**F**,**H**,**J**) represent the onset of MNV in the fellow eye. (**A**) B-scan with a spectral domain OCT image of the macula showed no retinal exudate. (**B**) B-scan with a swept-source OCT image of the macula showed steep PED with SRF above the pachyvessels (arrowhead). (**C**,**D**) Early-phase FA imaging showed an area of hyperfluorescence in the nasal and temporal superior macula. (**E**) Early-phase ICGA imaging showed pachyvessels corresponding to the area of hyperfluorescence on FA image. An area of hypofluorescence (arrow) was observed adjacent to choroidal vascular dilation. (**F**) Early-phase ICGA imaging showed abnormal vessels (arrow) in the area of hypofluorescence on ICGA prior to the onset of MNV. (**G**) Late-phase FA imaging showed an area of hyperfluorescence in the nasal and temporal superior macula. (**H**) Late-phase FA imaging showed the area of hyperfluorescence in the nasal and temporal superior macula. (**I**) Late-phase ICGA imaging showed an area of hypofluorescence near the pachyvessel. (**J**) Late-phase ICGA imaging showed areas of hyperfluorescence as polypoidal lesions (arrow).

**Table 1 jcm-13-05394-t001:** Clinical characteristics of patients with unilateral MNV with no drusen in the fellow eye.

	No-PHS (*n* = 29)	PHS (*n* = 29)	*p*
Age (years), mean (SD)	73.2 (8.2)	69.8 (9.6)	0.17
Sex (female), No. (%)	8 (27.6)	9 (31.0)	0.77
Hypertension, No. (%)	16 (55.2)	12 (41.4)	0.29
Diabetes, No. (%)	8 (27.6)	6 (20.7)	0.54
Smoking habits (ever-smokers), No. (%)	19 (65.5)	18 (62.1)	0.78
SFCT of the fellow eye (µm), mean (SD)	187.0 (84.8)	249.9 (124.4)	0.048
Presence of IRF, No. (%)	3 (10.3)	4 (13.8)	0.69
Presence of SRF, No. (%)	28 (96.6)	29 (100.0)	0.24
Presence of PPE, No. (%)	11 (37.9)	23 (79.3)	<0.001
Presence of polypoidal lesion, No. (%)	13 (44.8)	19 (65.5)	0.11
Presence of SHRM, No. (%)			0.036
Exudation	6 (20.7)	14 (48.3)	
Hemorrhage	7 (24.1)	8 (27.6)	
Neovascular tissue	2 (6.9)	0 (0.0)	
No SHRM	14 (48.3)	7 (24.1)	

SD, standard deviation; SFCT, subfoveal choroidal thickness; IRF, intraretinal fluid; SRF, subretinal fluid; PPE, pachychoroid pigment epitheliopathy; SHRM, subretinal hyper-reflective material.

**Table 2 jcm-13-05394-t002:** One-year outcome of patients with unilateral MNV receiving IVA monotherapy.

	No-PHS (*n* = 29)	PHS (*n* = 29)	*p*
Baseline, mean (SD)			
BCVA (logMAR)	0.23 (0.20)	0.35 (0.36)	0.21
CRT (µm)	318.7 (60.6)	317.8 (97.2)	0.41
SFCT (µm)	213.8 (94.0)	267.2 (120.4)	0.09
Final visit, mean (SD)			
BCVA (logMAR)	0.10 (0.18)	0.13 (0.29)	0.65
CRT (µm)	253.4 (69.2)	212.6 (38.1)	0.004
SFCT (µm)	188.8 (94.6)	231.6 (129.1)	0.36
Dry macula after loading dose, No. (%)	18 (62.1)	23 (79.3)	0.15
Number of injections, mean (SD)	7.0 (2.9)	5.4 (2.2)	0.039

SD, standard deviation; BCVA, best-corrected visual acuity; CRT, central retinal thickness; SFCT, subfoveal choroidal thickness.

## Data Availability

The data used to support the findings of this study are restricted by the Kawasaki Medical School Ethics Committee to protect patient privacy. Data are available from Hiroyuki Kamao [hironeri@med.kawasaki-m.ac.jp] for researchers who meet the criteria for access to confidential data.

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
