# Peer review of "Clinical Characteristics of Punctate Hyperfluorescence Spots in the Fellow Eye of Patients with Unilateral Macular Neovascularization with No Drusen"

_jcm, 2024, doi:10.3390/jcm13185394_

Round 1
Reviewer 1 Report
Comments and Suggestions for Authors
Dear authors,
I am really glad to review your interesting work on clinical assessment related to AMD patients with no drusen that can be very helpful further for diagnosis and treatment. Although the work defined and manuscript written is excellent, but some corrections/revisions needed before for final publishing. Attached are my comments for your review, that needs to be addressed.
-----------
Comments to the authors
I appreciate the opportunity to review this interesting report on assessing the clinical characteristics of patients with macular neovascularization (MNV) with no drusen. While the paper addresses an interesting issue, it needs some minor corrections followed by revision. Also, some minor editing of english language and proofreading is required throughout the manuscript.
My comments are below:
Abstracts: “Fellow eyes in the PHS group had a thicker choroid (p< 0.05) and higher prevalence of pachychoroid pigment epitheliopathy (PPE) (p < 0.001). MNV eyes in the PHS group had a thicker choroid (p = 0.09)”…
I couldn’t figure the differences in both the statements. If this can be checked.
Introduction: This needs to be bit elaborated more. Or some figures/diagram would help in describing PNV, CSC, PCV, PHS etc, better than just the text.
Figure 1: “A representative case of unilateral MNV with PHS in the fellow eye” Is the Figure 1A and 1B from one patient? Since both are color fundus photography how they are different.
Same goes for figure 2.
Line 156: “All patients” …. Do the authors say all 119 patients?
Line 191-194: “Among 119 eyes of the 119 consecutive patients with unilateral MNV with no drusen in the fellow eye, we excluded 61 eyes of 61 patients: 31 lacked ICGA data, 18 were treated with different methods, and 12 were followed up less than one year. A total of 58 eyes from 58 patients with treatment-naïve unilateral MNV were enrolled in our study”
Please check this. If the authors enrolled 58 patients then why 119 described. Also, if this paragraph can be included in Methods section to make it better.
Figure 4: Mention “Months” below X-axis
Figure 5 and 6: If the figures (both PHS and no-PHS) can be compared side wise, as the legends for the both the figures are similar in words and looks repeated.
Discussion: “Our study revealed that the PHS group had higher choroidal thickness in the eyes with MNV as well as fellow eyes, a higher prevalence of PPE, and a lower proportion of patients retreated with IVA therapy” ………. “The no-PHS group in this study had no drusen and lacked pachychoroid disease characteristics (thinner choroid, lower prevalence of PPE, and frequent anti-VEGF therapy”
Does this mean patient with PHS have higher possibility of diagnosis and treatment better the patient showing no PHS?
Line 285-286: “Our study revealed that in patients with unilateral MNV with no drusen in the fellow eye, fellow eyes in the PHS group” …Fellow eye looks repeated. Please check this.
Comments on the Quality of English LanguageProper proofreading and English language corrections needed during revision of the manuscript.
Author Response
Responses to the comments of Reviewer # 1
Thank you very much for reviewing our manuscript and offering helpful comments. We have carefully reviewed the comments and revised the manuscript accordingly. The following pages list our point-by-point responses to all the comments. Revisions in the text are highlighted for additions and strikethrough font (example) for deletions.
We added the following data:
- The description of the study's objective was in the Introduction section (lines 32-27).
- The description of pachychoroid diseases was in the Introduction section (lines 38-40 and lines 45-47).
- The results that demonstrate our anti-VEGF treatment methods (PRN and TAE regimen) were in Figures 1 and 2 (lines 105 and 121).
- Abstracts: “Fellow eyes in the PHS group had a thicker choroid (p< 0.05) and higher prevalence of pachychoroid pigment epitheliopathy (PPE) (p < 0.001). MNV eyes in the PHS group had a thicker choroid (p = 0.09)”…
I couldn’t figure the differences in both the statements. If this can be checked.
- Response: We apologize if this is an inaccurate response to your point. In this study, we assessed patients with unilateral MNV with no drusen in the fellow eye. The patients had MNV in one eye (MNV eye) and no drusen in the other eye (fellow eye). For example, in Figure 1, the right eye (Fig 1B) had MNV, and the left eye (Fig 1A) had no drusen (fellow eye). We evaluated the fellow eyes and classified the patients into two groups according to the presence or absence of PHS. The PHS group had a thicker SFCT of both the MNV eyes and fellow eyes than the no-PHS group. The mean SFCT of MNV eyes were 213.8 µm in the no-PHS group and 267.2 µm in the PHS group. Although this study included a relatively small sample size (each 29 patients), the p-value was 0.09; therefore, we considered that there were significant differences in SFCT.
- Introduction: This needs to be bit elaborated more. Or some figures/diagram would help in describing PNV, CSC, PCV, PHS etc, better than just the text.
- Response: Thank you very much for your suggestion. We added the following text in the Introduction section: In particular, CSC, PPE, PNV, and PCV have been speculated to represent a sequential disease process caused by pathological thick choroid [7] (lines 45-47).
- Dr. Yanagi (Reference 7) summarized CSC, PPE, PNV, and PCV. We evaluated patients with unilateral MNV with no drusen in the fellow eye, and we did not diagnose PCV, CSC, or PNV. Hence, we did not include a detailed description of them in this study.
- Regarding PHS, we added text showing the ICGA image of PHS (Figure 1G) in the Results section.
- Figure 1: “A representative case of unilateral MNV with PHS in the fellow eye” Is the Figure 1A and 1B from one patient? Since both are color fundus photography how they are different.
- Response: All images in Figure 1 were from the same patient. Figure 1A is the fellow eye (left eye), and Figure 1B is the MNV eye (right eye) (line 105).
- Same goes for figure 2.
- Response: All images in Figure 2 were from the same patient. Figure 2A is the fellow eye (right eye), and Figure 2B is the MNV eye (left eye) (line 121).
- Line 156: “All patients” …. Do the authors say all 119 patients?
- Response: Thank you very much for noting the ambiguous description. We revised “All patients” to “All enrolled patients” (line 177 and line 185).
- Line 191-194: “Among 119 eyes of the 119 consecutive patients with unilateral MNV with no drusen in the fellow eye, we excluded 61 eyes of 61 patients: 31 lacked ICGA data, 18 were treated with different methods, and 12 were followed up less than one year. A total of 58 eyes from 58 patients with treatment-naïve unilateral MNV were enrolled in our study”
Please check this. If the authors enrolled 58 patients then why 119 described. Also, if this paragraph can be included in Methods section to make it better.
- Response: We thank the reviewer for these comments. Your suggestion is correct. We moved the pointed sentences to the Methods section (lines 98-100).
- Figure 4: Mention “Months” below X-axis.
- Response: Thank you very much for your suggestion. We moved “Months” below X-axis (line 259).
- Figure 5 and 6: If the figures (both PHS and no-PHS) can be compared side wise, as the legends for the both the figures are similar in words and looks repeated.
- Response: We thank the reviewer for these comments. Figure 5 showed the development of MNV in the fellow eye of the patient with PHS, and Figure 6 showed the development of MNV in the fellow eye of the patient without PHS. We confirmed that the Legends properly explain the figures, even though the Legends are similar in words, because we presented the fundus photograph images, fundus angiography images, and OCT images in the same pattern.
- Discussion: “Our study revealed that the PHS group had higher choroidal thickness in the eyes with MNV as well as fellow eyes, a higher prevalence of PPE, and a lower proportion of patients retreated with IVA therapy” ………. “The no-PHS group in this study had no drusen and lacked pachychoroid disease characteristics (thinner choroid, lower prevalence of PPE, and frequent anti-VEGF therapy”
Does this mean patient with PHS have higher possibility of diagnosis and treatment better the patient showing no PHS?
- Response: Yes, it was. This study suggested that patients with no drusen + PHS in the fellow eyes have pachychoroid disease in the MNV eyes and respond better to anti-VEGF therapy.
- There are no diagnostic criteria for pachychoroid diseases, and pachychoroid diseases are diagnosed based on several clinical findings in the MNV eye, such as choroidal vascular hyperpermeability and pachyvessel. However, ​​large bleeding or PED could mask these characteristic clinical findings. Therefore, we believe that a more accurate diagnosis of pachychoroid can be made by the clinical findings of the fellow eye as well as that of the MNV eye.
- Line 285-286: “Our study revealed that in patients with unilateral MNV with no drusen in the fellow eye, fellow eyes in the PHS group” …Fellow eye looks repeated. Please check this.
- Response: We thank the reviewer for these comments. We confirmed that the Discussion properly explain our results (lines 306-308).
We look forward to hearing from you about our submission. We are happy to respond to any further questions and comments.
Sincerely yours,
Hiroyuki Kamao, MD, PhD
Department of Ophthalmology, Kawasaki Medical School, 577 Matsushima, Kurashiki, Okayama, 701-0114, Japan.
Tel: -81-8-6462-1111, Fax: -81-8-6464-1565
hironeri@med.kawasaki-m.ac.jp
Reviewer 2 Report
Comments and Suggestions for Authors
The manuscript is well-written and covers pachychoroid diseases effectively. However, I recommend that the authors clarify the study's objective and provide a figure illustrating the classification of the treatment groups and regimens.
Comments on the Quality of English LanguageGood English writing.
Author Response
Responses to the comments of Reviewer # 2
Thank you very much for reviewing our manuscript and offering helpful comments. We have carefully reviewed the comments and revised the manuscript accordingly. The following pages list our point-by-point responses to all the comments. Revisions in the text are highlighted for additions and strikethrough font (example) for deletions.
We added the following data:
- The description of the study's objective was in the Introduction section (lines 32-27).
- The description of pachychoroid diseases was in the Introduction section (lines 38-40 and lines 45-47).
- The results that demonstrate our anti-VEGF treatment methods (PRN and TAE regimen) were in Figures 1 and 2 (lines 105 and 121).
- The manuscript is well-written and covers pachychoroid diseases effectively. However, I recommend that the authors clarify the study's objective and provide a figure illustrating the classification of the treatment groups and regimens.
- Response: Thank you very much for your suggestion. We added the description of the study's objective, which is to assess predictive factors for the response of anti-VEGF therapy to reduce the burden of anti-VEGF therapy in the Introduction section (lines 32-37).
- We added the results that demonstrate our anti-VEGF treatment methods in Figures 1 and 2. Specifically, Figures 1I and 1J showed the PRN regimen in which no treatment was administered due to the absence of recurrence after the loading dose regimen. Figures 2I and 2J showed the TAE regimen in which anti-VEGF treatment was resumed upon recurrence after the loading dose regimen (lines 105 and 121).
We look forward to hearing from you about our submission. We are happy to respond to any further questions and comments.
Sincerely yours,
Hiroyuki Kamao, MD, PhD
Department of Ophthalmology, Kawasaki Medical School, 577 Matsushima, Kurashiki, Okayama, 701-0114, Japan.
Tel: -81-8-6462-1111, Fax: -81-8-6464-1565
hironeri@med.kawasaki-m.ac.jp